# Feasibility of Four Interventions to Improve Treatment Adherence in Migrants Living with HIV in The Netherlands

**DOI:** 10.3390/diagnostics10110980

**Published:** 2020-11-20

**Authors:** Sabrina K. Been, David A.M.C. van de Vijver, Jannigje Smit, Nadine Bassant, Katalin Pogány, Sarah E. Stutterheim, Annelies Verbon

**Affiliations:** 1Division of Infectious Diseases, Department of Internal Medicine, Erasmus University Medical Center, P.O. Box 2040, 3000 CA Rotterdam, The Netherlands; n.bassant@erasmusmc.nl; 2Viroscience Department, Erasmus University Medical Center, P.O. Box 2040, 3000 CA Rotterdam, The Netherlands; d.vandevijver@erasmusmc.nl; 3Department of Internal Medicine, Maasstad Hospital, P.O. Box 9100, 3007 AC Rotterdam, The Netherlands; SmitJ@maasstadziekenhuis.nl (J.S.); PoganyK@maasstadziekenhuis.nl (K.P.); 4Department of Work and Social Psychology, Faculty of Psychology and Neuroscience, Maastricht University, P.O. Box 616, 6200 MD Maastricht, The Netherlands; s.stutterheim@maastrichtuniversity.nl; 5Department of Medical Microbiology and Infectious Diseases, Erasmus University Medical Center, P.O. Box 2040, 3000 CA Rotterdam, The Netherlands

**Keywords:** HIV/AIDS, migrants, treatment adherence, interventions

## Abstract

We evaluated the feasibility and efficacy of four existing interventions to improve adherence to them in migrants living with HIV (MLWH): directly administered antiretroviral therapy (DAART), group medical appointments (GMA), early detection and treatment of psychological distress, and peer support by trained MLWH. At baseline and after the interventions, socio-demographic characteristics, psychosocial variables, and data on HIV treatment adherence were collected. The two questionnaires were completed by 234/301 (78%) MLWH included at baseline. Detectable HIV RNA decreased (from 10.3 to 6.8%) as did internalized HIV-related stigma (from 15 to 14 points), and self-reported adherence increased (between 5.5 and 8.3%). DAART and GMA were not feasible interventions. Screening of psychological distress was feasible; however, follow-up diagnostic screening and linkage to psychiatric services were not. Peer support for and by MLWH was feasible. Within this small intervention group, results on HIV RNA < 400 copies/mL (decrease of 23.6%) and outpatient clinic attendance (up to 20.4% kept more appointments) were promising.

## 1. Introduction

Since the introduction of combination Antiretroviral Therapy (cART) in 1996, HIV infection has become a manageable chronic disease with a greatly improved life expectancy [1,2]. However, despite advances in access to treatment and treatment regimens with fewer pills and fewer side effects, adherence to cART remains one of the main challenges to achieving optimal HIV treatment outcomes [2].

Several factors have been associated with adherence to cART, including clinical appointment attendance and psychosocial factors such as HIV-related stigma, social support, psychological distress, and quality of life [3,4,5,6]. Migrants living with HIV (MLWH), who, in Europe, comprised 40% of all people diagnosed with HIV between 2008 and 2017 [7], have worse psychosocial and clinical outcomes compared to people living with HIV (PLWH) born in Western Europe [8,9,10,11,12]. MLWH can thus be considered a “key population”. Specific risk factors for non-adherence to cART in MLWH include experiencing low social support, having low educational attainment, and experiencing low treatment adherence self-efficacy [13]. Because risk factors for non-adherence significantly influence treatment outcomes in MLWH, finding interventions that address risk factors for non-adherence to cART in this population is important.

Research on the feasibility and the efficacy of interventions aiming to improve adherence among MLWH is limited. We identified four existing interventions that could potentially improve treatment adherence in MLWH. The first was **directly administered antiretroviral therapy** (DAART). This intervention is derived from Directly Observed Therapy that has been used in tuberculosis control programs [14]. It usually entails patients taking their medications under the supervision of a healthcare provider multiple times per week alongside counseling on HIV treatment management during visits. The second intervention was **group medical appointments** (GMAs), in which multiple patients who share a medical condition collectively receive a consultation with their physician [15]. At these appointments, treatment adherence and living with the condition are discussed. Participants share their experiences of living with HIV, support each other, and learn from the experiences of other patients (peers) and the professional expertise of care providers (e.g., a nurse) [16]. GMA has been deployed in various healthcare settings and has been shown to improve knowledge about the condition and management of the condition, which can lead to improved self-management [17]. The third intervention was **screening and treatment of psychological distress** (i.e., anxiety and depression). Psychological distress has been found to contribute to non-adherence to cART, suboptimal treatment outcomes, and mortality [3,18,19,20], and previous studies have shown prevalence rates of 33% and 12.8–78% for anxiety and depression among PLWH [18,21,22]. Screening for psychological distress is thus important and subsequent treatment of psychological distress may be a key tool for increasing cART adherence. The fourth intervention was **peer support** by MLWH, whereby individuals have increased access to emotional support, informational support, and appraisal support [23,24]. Peer support has been recommended as a valuable addition to the clinical care of people living with chronic diseases, especially for individuals who are hard to reach due to individual (e.g., psychological distress) or demographic (e.g., ethnic minority) characteristics [25]. In low-income and middle-income countries, support from peers has shown positive effects on HIV treatment adherence and outcomes, and when compared to standard of care, was superior in reaching viral suppression [26,27,28].

The objective of the current study was to evaluate the feasibility and, if feasible, the efficacy of these four interventions. This study is the first to explore DAART, GMAs, screening and treatment of psychological distress, and peer support as interventions to improve HIV treatment adherence in a large cohort of non-Western MLWH in a Western European context.

## 2. Methods

### 2.1. Study Design and Context

This study is embedded in the ROtterdam ADherence (ROAD) project, which is a quasi-experimental intervention study in which the primary aim was to increase adherence in first- and second-generation adult immigrant people living with HIV (The Netherlands National Trial Registry Number—NTR4941). In the ROAD project, we prospectively followed a large group of MLWH over a two to three year period. In the study reported here, we explored the feasibility and, if feasible, the efficacy of four interventions that aim to improve adherence to HIV treatment. Our research questions were: (1) “Are DAART, GMA, screening and treatment of psychological distress, and peer support by MLWH feasible interventions in MLWH, living in The Netherlands?” and (2) “Among feasible interventions, to what extent are the interventions efficacious in improving HIV treatment adherence?”. The Medical Ethics Committee of the Erasmus University Medical Center and Maasstad Hospital have evaluated and approved the study on 28 September 2012 (reference number: MEC-2012-399).

### 2.2. Sampling and Recruitment

Participants were included if they originated from a region outside of Western Europe (first or second generation), were aged 18 years or older, were diagnosed with HIV, and were sufficiently fluent in Dutch, English, French, Spanish, or Portuguese. Between November 2012 and July 2013, a total of 352 PLWH from outpatient clinics of two Rotterdam HIV treatment centers (Erasmus University Medical Center and Maasstad Hospital) participated in the study [13].

### 2.3. Selection for Intervention(s)

Participants who experienced substantial difficulty living with HIV, had problems with HIV treatment adherence, or were at risk for non-adherence despite adequate access to HIV care were allocated to an intervention by a multidisciplinary HIV care team who determined which intervention (DAART, GMA, or peer support) or combination of interventions would be most appropriate for each given patient. All participants were asked to complete the Hospital Anxiety and Depression Scale (HADS), and allocation to the early detection and treatment of psychological distress intervention was made based on participants’ scores on the HADS [29]. We allowed allocation to multiple interventions because adherence to antiretroviral therapy is influenced by multiple factors, and multiple interventions have been shown to yield better adherence than single interventions [28].

Directly Administered Antiretroviral Therapy: The DAART intervention required participants to attend three to five additional appointments per week (Monday–Friday) with a nurse specialized in HIV care at an outpatient clinic where they would take their cART under supervision and where they could discuss successes and challenges in cART adherence. Other doses were to be self-administered. After six months, participants were expected to attend one to two visits per week and then, later, monthly visits, with a total duration of the intervention being up to 12 months. Patients’ travel costs to the hospital were reimbursed.

Group Medical Appointment: For the GMAs, eligible participants were initially approached by their physician or HIV nurse during their regular consultation or by phone, after which they received a patient information leaflet about the GMA. Prior to participation, patients signed a confidentiality form stating that they would not discuss patient-related information that was discussed during the GMA. We aimed to include a demographically heterogeneous group of PLWH in the GMAs because previous experience had shown that creating a heterogeneous group seemed to be favorable for group processes [30].

Screening and Treatment of Psychological Distress: All MLWH participating in the ROAD project were asked to complete the 14-item Hospital Anxiety and Depression Scale (HADS) [29], which is a widely used screening tool for psychological distress in multiple languages [31]. A HADS score of ≥15 (HADS+) indicates psychological distress [32,33,34], and all participants with a HADS+ were subsequently invited to complete the Composite International Diagnostic Interview (CIDI), which is a questionnaire designed to assess psychological disorders according to the definitions and criteria in the ICD-10 and DSM-IV [35,36,37]. When the CIDI results indicated psychological distress and the patient was not already enrolled in psychiatric care, a consultation with a psychiatrist was offered and the treating HIV physician was informed about the results (care as usual). This process has been described in more detail elsewhere [38].

Peer Support: ROAD participants eligible for peer support were approached by their HIV care provider. When they consented to participate, the HIV care provider referred them to Stichting Mara (informal care organization), who matched them to a trained peer MLWH. Peers were MLWH who had a history of being adherent to cART and/or outpatient clinic appointments. Previous research has shown that peers are best when matched on similar characteristics like demographics (e.g., sex, age, country of origin), behaviors (e.g., substance abuse), and conditions (e.g., living with HIV) [39]. Peers were selected by their HIV care providers and, prior to being matched, received a three-day training in which they were provided with a medical update on HIV and cART and learned how to deal with psychological distress among PLWH, HIV-related stigma, and taboos. They also discussed setting appropriate limits in the peer relationship, when to refer matched patients to formal care, and their own motivations for participation. Peers participated on a voluntary basis and received ongoing support from the counselors from Stichting Mara.

### 2.4. Data Collection

Prior to data collection, written informed consent was provided by all participants. Methods for baseline data collection of sociodemographic and psychosocial variables have been presented previously [13]. In short, data were collected via medical records and an interviewer-administered questionnaire. Assessment of psychosocial variables occurred via the eight-item Medical Outcomes Study Social Support Survey (mMOS-SS) [40], the six-item Internalized AIDS-Related Stigma Scale [41], the 12-item HIV Treatment Adherence Self-Efficacy Scale (HIV-ASES) [42], the 12-item Short-Form Health Survey (SF-12) [43], and the Hospital Anxiety and Depression Scale (HADS, only at baseline) [29]. Three four-item measures developed from adherence questions used in previous studies were used to measure self-reported adherence to cART [13]. The results of all three developed measures presenting stricter (Measure I) and less strict (Measure II and III) criteria for adherence are presented. Approximately 1–2.5 years after inclusion, participants were again approached during their outpatient clinic appointment to complete this questionnaire (T1) again.

HIV RNA values, CD4 cell counts, and information on cART use were collected from the ATHENA national observational HIV cohort database (the Dutch national HIV registry of HIV treatment centers). When incomplete, data were cross-checked with medical records. HIV RNA and outpatient clinic attendance data on participants in the peer support intervention were collected between one year before the match and up to two years afterward.

Additionally, participants in the GMA were asked to anonymously complete an evaluation form that included questions on how they experienced the GMA and whether they would participate again.

### 2.5. Data Analyses

Sociodemographic, clinical, and outpatient clinic attendance data were processed in Microsoft Office Excel and SPSS Statistics 24.0 (IBM New York, NY, USA. The Wilcoxon matched-pairs test, paired-samples *t*-test, McNemar test, and McNemar-Bowker test were used to compare data of the 2 questionnaires of the participants who used cART ≥ 6 months at the time of inclusion.

## 3. Results

### 3.1. Participant Characteristics

Characteristics of the included population in the ROAD project have been reported elsewhere [13]. In short, 352 of the 857 eligible MLWH were included at baseline (41.1% of eligible MLWH) (Figure 1). Approximately 58% were men, almost all were first-generation immigrants (94.6%), and the majority originated from Sub-Saharan Africa (40.6%), followed by Latin America (22.7%), the Caribbean (19.6%), and then other regions (17.0%). Of these 352 participants, 301 used cART ≥ 6 months prior to inclusion (cART-experienced) (Figure 1). A total of 234/301 of the cART-experienced participants completed both the first and the second questionnaire (baseline and T1) (response = 78%) (Figure 1). The main reasons for not completing T1 were of a logistical nature (*n* = 44, 66%) and refusal to participate (*n* = 17, 25%).

Table 1 presents characteristics of the 234 cART-experienced participants who completed both questionnaires. Overall, the percentage of participants with detectable HIV RNA (≥50 copies/mL) decreased (10.3% to 6.8%, *p* = 0.12), internalized HIV-related stigma score decreased (15 to 14, *p* < 0.01), and the percentage of participants who were adherent increased (between 5.5% and 8.3%, *p*-values between 0.15 and <0.05).

### 3.2. Feasibility of the Interventions

Table 2 presents a summary of the interventions’ descriptions, feasibility, and limitations.

Directly Administered Antiretroviral Therapy (DAART): Unfortunately, no study participants were willing to take part in this intervention. They objected to the necessary investment of time for outpatient clinic visits. Furthermore, multiple study participants indicated to their HIV care provider that they found the intervention intrusive and impractical. Therefore, we ascertained that this intervention is currently not feasible for MLWH in The Netherlands.

Group Medical Appointment: Four GMAs were held, with a total of 27 participants, including one person participating two times. GMAs were open to all PLWH including MLWH. In each GMA, six to nine PLWH participated. HIV care providers found it difficult to find patients willing to participate in a GMA, especially MLWH. HIV care providers indicated that this was because many patients had concerns about revealing their HIV status to others and/or discussing their medical situation within a group. They feared third party disclosure and stigmatization. In addition, a number of eligible MLWH were not able to participate because the GMAs were held in Dutch and they did not speak Dutch. Of the 27 GMA participants, 13 were included in the ROAD project. While the threshold to participate in the GMA seemed to be high, participants were very positive. The mean overall rating given to the GMAs by participants was 8.2 out of 10. Additionally, 67% of the 27 participants would choose a GMA for their next outpatient clinic appointment, 78% felt better informed about HIV compared to an individual consultation, 93% indicated having learned from the questions asked by other PLWH, and 96% would recommend it to others. Unfortunately, we were not able to ascertain satisfaction specifically among MLWH, as the evaluation form was completed anonymously. Nonetheless, in light of the challenges experienced with participant inclusion, we conclude that intervention is not highly feasible in MLWH.

Screening and Treatment of Psychological Distress: In total, 306 of the 352 participants (87%) completed the HADS and 106 (35%) had a HADS+ indicating psychological distress [38]. The HADS thus seems to be feasible for administration in a multilingual population. However, the HADS only provides an indication of psychological distress in the past seven days and therefore should be followed up upon quickly. As previously reported [38], following up on participants with a HADS+ and having them complete the CIDI was challenging. The CIDI was completed by only 60 of the 106 (58%) eligible participants with a HADS+. This is possibly because it was only possible to administer the CIDI in Dutch and English. Follow-up was further hampered by participants refusing to complete the CIDI or not attending the appointment to do the CIDI, as well as other logistical challenges. We therefore conclude that the feasibility of administering the CIDI to a group of MLWH previously screened with the HADS is minimal. Among those who did complete the CIDI, 21 received a diagnosis of depression or anxiety disorder [38], six of whom were already enrolled in psychiatric care. Unfortunately, of the remaining 15 participants, seven either refused an appointment with a psychiatrist or did not attend the scheduled appointment. In the end, only eight participants met with a psychiatrist. Therefore, we were not able to determine the efficacy of this intervention.

Peer Support: Within the ROAD project, 23 MLWH predominantly originating from Sub Saharan Africa, the Caribbean, and Latin America (additional socio-demographic characteristics are presented in Appendix A) received peer support, and more than 20 peers received training to provide peer support. Not all participants to whom this intervention was offered agreed to participate, mostly due to fear of third party disclosure. Most matched patients found it pleasant to have a peer with a similar background (e.g., gender and/or country of origin and/or language), but some participants requested a peer with a Dutch country of origin, also because they feared third-party disclosure. 

### 3.3. Efficacy of the Interventions

Peer Support: Of the 17 MLWH who were cART-experienced (six were cART-naïve or used cART < 6 months prior to inclusion), eight (47%) had an undetectable HIV RNA (<50 copies/mL) in the year before they were matched with a peer (Table 3). One and two years after the intervention, HIV RNA was undetectable in 10 (59%) (Table 3). The percentage of patients with HIV RNA < 400 copies/mL increased from 53% before the match to 77%, one and two years later. The percentage of patients who attended all outpatient clinic appointments increased from 48% one year before the match to 50% one year later and 68% two years later.

Overall Effectiveness of the Interventions: In all patients who completed both the baseline and T1 questionnaires, detectable viral load (HIV RNA > 50 copies/mL) decreased from 10.2% to 6.8% (*p* = 0.12) (Table 1). Self-reported adherence percentages increased in the more conservative Measure I (34.6% to 41.0%, *p* < 0.10) and Measure II (64.2% to 51.7%, *p* = 0.15), and in the less conservative Measure III (49.6% to 57.3%, *p* < 0.05). The score for internalized HIV-related stigma decreased significantly (*p* < 0.05) between baseline and T0.

## 4. Discussion

In this study, we followed 234 MLWH over a period of approximately 2.5 years and assessed the feasibility, and, if feasible, the efficacy of directly administered antiretroviral therapy (DAART), group medical appointment (GMA), screening and treatment of psychological distress, and peer support for and by MLWH on improving HIV treatment adherence among MLWH. The results demonstrate that DAART and GMA are currently not feasible interventions within an MLWH population living in Western Europe. In addition, while screening of psychological distress with the HADS was feasible in this population, follow-up screening with the CIDI and subsequent linkage to psychiatric care services were not. Peer support for and by MLWH appeared to be a feasible intervention, and the results within a small intervention group on viral load and outpatient clinic attendance were promising. 

Our finding that DAART was not feasible is in line with some literature showing poor feasibility for DAART interventions with other key populations of PLWH with a higher risk for non-adherence, such as people engaged in substance use and people experiencing homelessness [44,45]. Although we were aware of existing evidence that DAART had not worked for other key populations, we nonetheless wanted to offer this intervention to MLWH as MLWH may differ from the key populations in which DAART was previously applied. However, in our study, the participants found the intervention intrusive and impractical, which has also been described elsewhere [46]. It is possible that in another context, e.g., the participant’s home, this intervention would be more feasible among MLWH as this would obviate the need for the participant to travel to the clinic. Future studies should investigate the feasibility of this adapted version of the intervention among MLWH.

Our findings also showed that there were challenges in finding PLWH willing to participate in a GMA. At the same time, those who did participate appreciated the intervention. Similar to the findings in our study, Smets and Zantinge et al. both described recruitment challenges for GMAs and subsequent appreciation by those who did participate [30,47]. It thus appears that the feasibility of GMA in a population of MLWH is limited at this time. Fear of HIV-related stigma resulting from being recognized by other GMA participants and fear of third party disclosure appear to be key barriers to participation in GMAs for MLWH. Future research should investigate the role of HIV-related stigma of MLWH’s willingness to participate in GMA interventions. We contend that HIV-related stigma should first be addressed before GMAs can be used for improving treatment adherence among MLWH.

In line with our expectations, our findings show that the HADS is a relatively feasible tool to administer in an outpatient clinical setting with a multicultural and multilingual population. Our findings also show poor feasibility for the administration of the CIDI as there were significant challenges in getting participants with a HADS+ to complete the CIDI. It is possible that CIDI completion rates may have been impeded by the fact that the CIDI is not available in multiple languages. This has been indicated earlier by Schrier et al. [48]. We therefore suggest that when psychological care is indicated, a diagnostic assessment be used that takes cultural differences into account.

Peer support seems to be the most feasible intervention that was tested in this study. Despite the small sample size, the results on HIV RNA and outpatient clinical appointments were promising. Similar results were found in a previously published systematic review and in a network meta-analysis performed in middle- and low-income countries [26,28]. Both Kanters et al. and Mills et al. found positive effects of peer support on HIV treatment adherence. In addition, when compared to standard care, peer support interventions were found to be superior in reaching viral suppression [28]. Mixed results were shown in two peer intervention studies performed in the United States of America [49,50]. This was partially attributed to the relatively short duration of the interventions (less than three months). Additionally, a qualitative study conducted with participants in our peer support project showed that peer support was reported to have increased patients’ knowledge about (living with) HIV, helped patients to feel emotionally supported, reduced perceived HIV-related stigma, and reduced disclosure concerns [51]. Additionally, that study also showed that the peers found the support they provided meaningful and a source of personal growth. Furthermore, the HIV care providers involved in this study saw a need for this intervention as multiple requests were made to match PLWH who were eligible but not enrolled in the ROAD project. We therefore believe that peer support for and by MLWH to improve HIV treatment adherence has potential, particularly for MLWH who originate from collectivist cultures where peer interaction and support provision within communities is highly valued. We recommend that future studies investigate this intervention on a larger scale than was possible in this study to better determine the effect on psychosocial and clinical outcomes.

This study has a number of strengths and limitations. One key strength is that we successfully included and followed up on 234 cART-experienced MLWH, which is a difficult-to-reach key population. To our knowledge, this is the largest sample of MLWH recruited for a study on testing the feasibility and possible efficacy of interventions aiming to improve HIV treatment adherence in Europe to date [28]. A second strength is that, rather than testing a single intervention, we used our resources to simultaneously test the feasibility and possible efficacy of multiple existing interventions in this specific sample. A limitation is that the sample size in the peer support intervention was relatively small, making the analyses possibly quite limited. Future studies should be designed with sufficient power to be able to determine the effectiveness of peer support. A second limitation is that we did not have a control group, e.g., MLWH who received standard care. However, we performed a pilot study aiming to determine the feasibility and possible efficacy of four interventions aimed to improve treatment adherence. In that pilot study, peer support was a feasible intervention. Nonetheless, future studies should use an intervention and control group to test the efficacy of this intervention. Therefore, we were not able to demonstrate that the overall differences found in detectable HIV RNA, experienced HIV-related stigma, and self-reported adherence at baseline and T1 were due to the interventions that were administered. It is possible that these results are, to some extent, the consequence of standard care or, alternatively, the Hawthorne effect [52]. It is possible that the focus on HIV treatment adherence with the administration of the baseline questionnaire for study inclusion made study participants (perhaps unconsciously) more aware of their HIV treatment adherence behavior and made health care providers more keen to encourage treatment adherence. However, we maintain that being offered and participating in several interventions to improve adherence likely changed adherence behavior, particularly in MLWH, many of whom had not or only selectively disclosed their HIV status to others.

In their systematic review, Whembolua et al. [53] found that disclosure of HIV status is related to decreased anxiety and emotional relief, increased social support, decreased high-risk sexual behaviors, increased help- and information-seeking behaviors, and, importantly, increased adherence to cART. A final limitation is that we did not use focus groups with the MLWH population to ascertain the acceptability of the interventions before implementing the interventions. Future studies aiming to test interventions should ideally gauge the acceptance of proposed interventions in the MLWH population beforehand. This can be done by organizing focus groups or by including key figures from the migrants’ communities in the research team, using a community-based participatory research approach [54,55].

In conclusion, our findings provide evidence for the feasibility, or lack thereof, of DAART, GMA, screening and treatment of psychological distress, and peer support for and by MLWH as interventions improving HIV treatment adherence. Of the four interventions, peer support appeared to be a feasible intervention, and the results on HIV RNA and outpatient clinic visit attendance were promising. Future studies should more extensively, and in a larger sample, evaluate peer support to further determine the effect on both psychosocial and clinical outcomes. Additionally, continuing efforts to assess interventions that aim to improve HIV treatment adherence among MLWH are recommended. In doing so, we recommend including key figures from migrant (living with HIV) communities in the development of such interventions to ensure that interventions are culturally sensitive and likely to be accepted by MLWH.

## Figures and Tables

**Figure 1 diagnostics-10-00980-f001:**
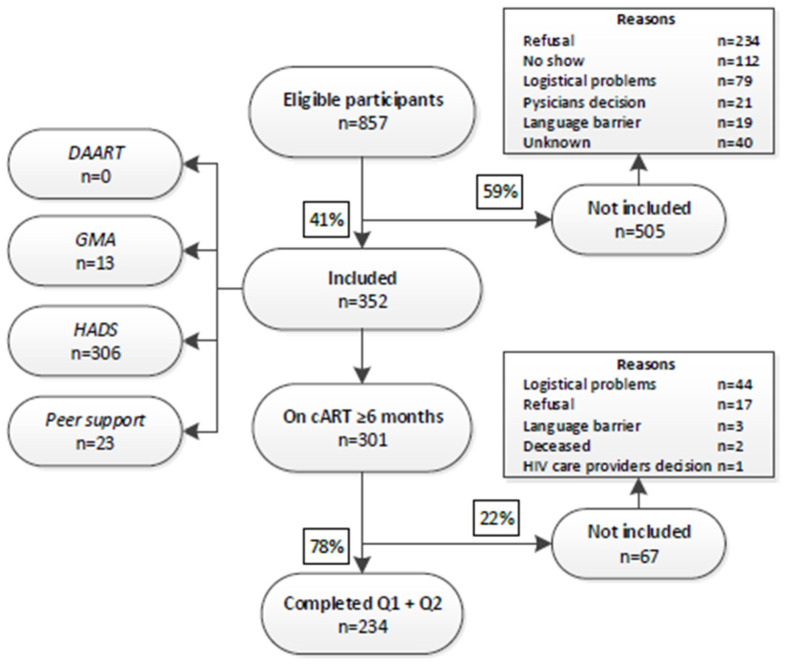
Patient inclusion ROtterdam ADherence (ROAD) Project. DAART, Directly Administered Antiretroviral Therapy; GMA, group medical appointments; HADS, Hospital Anxiety and Depression Scale.

**Table 1 diagnostics-10-00980-t001:** Sociodemographic characteristics and psychosocial variables.

Variable		Baseline *n* = 234	T1 *n* = 234	*p*
**HIV-RNA > 50 copies/mL (%)**		24 (10.3)	16 (6.8)	0.12 ^d^
**Age (mean, SD)**		43.2 (9.9)		
**Male gender (%)**		127 (54.3)		
**1st generation immigrant (%)**		225 (96.2)		
**Region of origin (%)**				
*Sub Saharan Africa*		103 (44.0)		
*Caribbean*		42 (17.9)		
*Latin America*		45 (19.2)		
*Other*		44 (18.8)		
**Sexual orientation (%) ^a^**				
*Heterosexual*		153 (65.4)		
*Gay/Bisexual*		71 (30.3)		
*Does not know*		7 (3.0)		
**Living situation (%)**				0.23 ^e^
*With family*		89 (38.0)	80 (34.2)	
*Alone*		85 (36.3)	96 (41.0)	
*Single parent*		45 (19.2)	44 (18.8)	
*Other*		15 (6.4)	14 (6.0)	
**Children (%) ^a^**		143 (61.1)		
**Educational attainment (%) ^a^**				
*No formal education/Primary school*		62 (26.5)		
*Secondary school*		69 (29.5)		
*Higher vocational school*		54 (23.1)		
*University*		47 (20.1)		
**Employment status (%)**				0.23 ^e^
*Paid employment*		92 (39.3)	88 (37.9)	
*Unemployed*		67 (28.6)	62 (26.5)	
*On sick leave*		26 (11.1)	25 (10.7)	
*Other*		49 (20.9)	59 (25.2)	
**Alcohol (%) ^a^**				
*Alcohol in the past 30 days*		130 (55.6)	122 (51.1)	0.37 ^d^
*Alcohol use ≥ 3 days/week*		40 (17.1)	35 (15.0)	0.42 ^d^
**Drugs (%) ^a^**				
*Drugs in the past 30 days*		39 (16.7)	36 (15.34)	0.72 ^d^
*Drugs use ≥ 3 days/week*		24 (10.3)	23 (9.8)	1.00 ^d^
**Psychosocial variables (median, IQR) ^a,b^**				
*Social support*		75 (40.6–90.6)	68.8 (43.8–87.5)	0.97 ^f^
*Int. HIV-related stigma*		15 (12–19)	14 (11–18)	<0.01 ^f^
*Adherence self-efficacy*		105.5 (91.8–116.0)	106 (89.8–115.0)	0.93 ^f^
*Quality of life (physical)*		51.3 (40.7–56.2)	50.4 (41.7–55.6)	0.36 ^f^
*Quality of life (mental)*		48.9 (38.9–57.2)	48.8 (37.8–56.3)	0.84 ^f^
**Self-reported adherence ^a,b^**				
*Measure I ^c^*	*Adherent*	81 (34.6)	96 (41.0)	0.07 ^d^
	*Non-adherent*	151 (64.5)	135 (57.7)	
*Measure II ^c^*	*Adherent*	108 (46.2)	121 (51.7)	0.15 ^d^
	*Non-adherent*	124 (53.0)	110 (47.0)	
*Measure III ^c^*	*Adherent*	116 (49.6)	134 (57.3)	<0.05 ^d^
	*Non-adherent*	116 (49.6)	97 (41.5)	

^a^ Missing values baseline questionnaire: sexual orientation (*n* = 3), children (*n* = 1), educational attainment (*n* = 2), alcohol (*n* = 1), drugs (*n* = 1), social support (*n* = 14), internalized HIV-related stigma (*n* = 11), adherence self-efficacy (*n* = 36), physical quality of life (*n* = 5), mental quality of life (*n* = 5), self-reported adherence (*n* = 2). ^b^ Missing values T1: alcohol (*n* = 3), drugs (*n* = 3), social support (*n* = 4), internalized HIV-related stigma (*n* = 8), adherence self-efficacy (*n* = 32), physical quality of life (*n* = 8), mental quality of life (*n* = 8), self-reported adherence (*n* = 3). ^c^ Measures present stricter (Measure I) and less strict (Measure II and III) criteria for adherence [13]. ^d^ McNemar test; ^e^ McNemar–Bowker test. ^f^ Paired—Samples *t* test.

**Table 2 diagnostics-10-00980-t002:** Summary of interventions’ descriptions, feasibility, and limitations.

Intervention	In Short	Feasible?	Inclusion Limitations
DAART	Supervised cART intake during additional outpatient clinic appointments	no	• necessary time investment • eligible participants found it intrusive and impractical
GMA	Shared medical consultation with peers	no	Fears for: • disclosure of HIV status • third party disclosure • stigmatization
HADS and CIDI	Screening and treatment of psychological distress	HADS—yes CIDI—no	In administering the CIDI: • language barriers • refusal to complete the questionnaire • logistical challenges For following up with psychiatrist: • refusing the appointment • no show
Peer support	Support by migrants living with HIV	yes	Fears for third party disclosure

DAART, Directly Administered Antiretroviral Therapy; GMA, group medical appointments; HADS, Hospital Anxiety and Depression Scale; CIDI, Composite International Diagnostic Interview.

**Table 3 diagnostics-10-00980-t003:** HIV RNA and outpatient clinic attendance.

Variable	Match − 1 year	Match + 1 year	Match + 2 years
**HIV-RNA ^a^**	(*n* = 17)	(*n* = 17)	(*n* = 17)
All values < 50 copies/mL (%)	8 (47.1)	10 (58.8)	10 (58.8)
All values < 400 copies/mL (%)	9 (52.9)	13 (76.5)	13 (76.5)
**Outpatient clinic attendance**	(*n* = 23)	(*n* = 22) ^b^	(*n* = 22) ^b^
Attended all appointments (%)	11 (47.8)	11 (50.0)	15 (68.2)

^a^ On cART < 6 months prior to match. ^b^ Lost to follow up, *n* = 1.

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
