# Peer review of "Feasibility of Four Interventions to Improve Treatment Adherence in Migrants Living with HIV in The Netherlands"

_diagnostics, 2020, doi:10.3390/diagnostics10110980_

Round 1
Reviewer 1 Report
-Please revise Figure1 to illustrate how many participants in each intervention
-Please provide table to compare results from each intervention. Even though DAART and GMA were not feasible interventions, it would help the reader to understand differences and limitation of each intervention.
-Please revise the conclusion and suggest how to apply each intervention in MLWH
Author Response
Concerning: Revision manuscript: "Feasibility of four interventions to improve treatment adherence in migrants living with HIV in the Netherlands"
Dear Ms. Tomovic,
We would like to thank the reviewers for their time and useful contribution that has helped to improve the manuscript. In addition, we are pleased to read that reviewer 2 mentions that the subject of our manuscript and its study population are considered as highly relevant for care providers managing people living with HIV.
Hereby, we would like to submit our revised manuscript for publication as an original article in Diagnostics after carefully evaluating the comments of reviewers 1, 2, 3, and revising our manuscript accordingly. We will respond point-by-point to the reviewers` comments.
Replies to the comments of reviewer 1.
- Please revise Figure1 to illustrate how many participants in each intervention
Author`s response:
We thank the reviewer for this comment and have revised Figure 1. It now includes the number of participants per intervention: DAART:0, GMA: 13, HADS: 306, and Peer support: 23.
- Please provide table to compare results from each intervention. Even though DAART and GMA were not feasible interventions, it would help the reader to understand differences and limitation of each intervention.
Author`s response:
We agree that an overview of the interventions, their feasibility, and limitations would be a good contribution to this paper. An additional table (Table 2) has been included and mentioned in paragraph ‘Feasibility of the interventions’ (page 8, line 222).
- Please revise the conclusion and suggest how to apply each intervention in MLWH
Author`s response:
We thank the reviewer for this comment. In the discussion section we have outlined how each intervention can be applied, what could be adapted or what should be investigated before the intervention can be tested again. This can be found on pages 11-14 (lines 303-306, 313-316, 322-324, and 340-345). In addition, we believe that the implementation of interventions among MLWH depends on the local context as MLWH populations and available services may vary. A good example of a method that takes this context into consideration, is community-based participatory research where key figures from the target populations` community are included in the research team. We commented about this in the Discussion (page 14, lines 373-376).

Reviewer 2 Report
The issue of adherence is critical in achieving viral suppression and HIV epidemic control. The author addresses therefore an important topic for service providers managing people living with HIV. The study population is also an important group to investigate because of migration related specific issues including socio-cultural disparities, low education level, communication challenges and stigma. Interventions selection was driven by a multidisciplinary team but current approaches have been more and more client centered to improve adherence and retention. Multiple interventions were mentioned but not described and therefore it was difficult to identify their acceptability, feasibility and impact. The sample size for the intervention that was feasible was very low. The lack of feasibility of the two other interventions and methodology shortcomings weakens the quality of the research.
The paper could focus on stigma and early detection and treatment of psychological distress which is very relevant and could be explored with more detailed analysis.
Author Response
Concerning: Revision manuscript: "Feasibility of four interventions to improve treatment adherence in migrants living with HIV in the Netherlands"
Dear Ms. Tomovic,
We would like to thank the reviewers for their time and useful contribution that has helped to improve the manuscript. In addition, we are pleased to read that reviewer 2 mentions that the subject of our manuscript and its study population are considered as highly relevant for care providers managing people living with HIV.
Hereby, we would like to submit our revised manuscript for publication as an original article in Diagnostics after carefully evaluating the comments of reviewers 1, 2, 3, and revising our manuscript accordingly. We will respond point-by-point to the reviewers` comments.
Replies to the comments of reviewer 2.
- Interventions selection was driven by a multidisciplinary team but current approaches have been more and more client centered to improve adherence and retention.
Author`s response:
We agree that including representatives from the target population in the research or project team is likely to improve adherence to and retention in interventions. In the ‘Discussion’ section we recommend to use this community-based participatory approach in future studies (page 14, lines 373-376).
- Multiple interventions were mentioned but not described and therefore it was difficult to identify their acceptability, feasibility and impact.
Author`s response:
A description of the rationale for the interventions has been given in the ’Introduction’ (pages 3-4, lines 65-89):
The first was directly administered antiretroviral therapy (DAART). This intervention is derived from Directly Observed Therapy that has been used in tuberculosis control programs (14). It usually entails patients taking their medications under the supervision of a healthcare provider multiple times per week alongside counseling on HIV treatment management during visits. The second intervention was group medical appointments (GMAs) in which multiple patients who share a medical condition collectively receive a consultation with their physician (15). At these appointments, treatment adherence and living with the condition are discussed. Participants share their experiences of living with HIV, support each other, and learn from the experiences of other patients (peers) and the professional expertise of care providers (e.g. a nurse) (16). GMA has been deployed in various healthcare settings and has been shown to improve knowledge about the condition, and management of the condition, which can lead to improved self-management (17). The third intervention was screening and treatment of psychological distress (i.e., anxiety and depression Psychological distress has been found to contribute to non-adherence to cART, suboptimal treatment outcomes, and mortality (3, 18-20), and previous studies have shown prevalence rates of 33% and 12.8%-78% for anxiety and depression among PLWH (18, 21, 22). Screening for psychological distress is thus important and subsequent treatment of psychological distress may be a key tool for increasing cART adherence. The fourth intervention was peer support by MLWH whereby individuals have increased access to emotional support, informational support, and appraisal support (23, 24). Peer support has been recommended as a valuable addition to the clinical care of people living with chronic diseases, especially for individuals who are hard to reach due to individual (e.g., psychological distress) or demographic (e.g., ethnic minority) characteristics (25). In low-income and middle-income countries, support from peers has shown positive effects on HIV treatment adherence and outcomes, and when compared to standard of care, was superior in reaching viral suppression (26-28).
A description of the practical implementation has been given in the ‘Methods’ under ‘Selection for the intervention(s)’ (pages 5-7, lines 132-168):
Directly administered antiretroviral therapy: The DAART intervention required participants to attend three to five additional appointments per week (Monday-Friday) with a nurse specialized in HIV care at an outpatient clinic where they would take their cART under supervision and where they could discuss successes and challenges in cART adherence. Other doses were to be self-administered. After six months, participants were expected to attend one to two visits per week and then, later, monthly visits, with a total duration of the intervention being up to 12 months. Patients’ travel costs to the hospital were reimbursed.
Group medical appointment: For the GMAs, eligible participants were initially approached by their physician or HIV nurse during their regular consultation or by phone, followed by a patient information leaflet about the GMA. Prior to participation, patients signed a confidentiality form stating that they would not discuss patient-related information that was discussed during the GMA. We aimed to include a demographically heterogeneous group of PLWH in the GMAs because previous experience had shown that creating a heterogeneous group seemed to be favorable for group processes (30).
Screening and treatment of psychological distress: All MLWH participating in the ROAD project were asked to complete the 14-item Hospital Anxiety and Depression Scale (HADS) (29), which is a widely used screening tool for psychological distress in multiple languages (31). A HADS score of ≥15 (HADS+) indicates psychological distress (32-34) and all participants with a HADS+ were subsequently invited to complete the Composite International Diagnostic Interview (CIDI), which is a questionnaire designed to assess psychological disorders according to the definitions and criteria in the ICD-10 and DSM-IV (35-37). When the CIDI results indicated psychological distress and the patient was not already enrolled in psychiatric care, a consultation with a psychiatrist was offered and the treating HIV physician was informed about the results (care as usual). This process has been described in more detail elsewhere (38).
Peer support: ROAD participants eligible for peer support were approached by their HIV care provider. When they consented to participate, the HIV care provider referred them to Stichting Mara (informal care organization) who matched them to a trained peer MLWH. Peers were MLWH who had a history of being adherent to cART and/or outpatient clinic appointments. Previous research has shown that peers are best when matched on similar characteristics like demographics (e.g., sex, age, country of origin), behaviors (e.g., substance abuse), and conditions (e.g., living with HIV) (39). Peers were selected by their HIV care providers and, prior to being matched, received a three day training in which they were provided with a medical update on HIV and cART, and learnt how to deal with psychological distress among PLWH, HIV-related stigma, and taboos. They also discussed setting appropriate limits in the peer relationship, when to refer matched patients to formal care, and one’s own motivations for participation. Peers participated on a voluntary basis and received ongoing support from the counselors from Stichting Mara.
- The sample size for the intervention that was feasible was very low.
Author`s response:
Indeed, the sample size of the peer support intervention was relatively small making the analyses possibly quite limited. We mentioned this in the ‘Discussion’ section (page 13, lines 351-354).
The results of this study do contribute to a better understanding of the feasibility of adherence-improving interventions among MLWH. Current research on the effect of peer support on treatment adherence is minimal, inconclusive, and usually does not specifically target MLWH living in Europe. The results of this pilot study are thus important first steps in addressing these gaps in the literature. However, a more extensive evaluation is needed. This could be done in a full trial with an intervention and control cohort and both quantitative and qualitative research components. In addition, the included sample should have sufficient power and, because we can expect challenges with regard to patient inclusion, it is recommended that more than two HIV treatment centers participate in this trial.
- The lack of feasibility of the two other interventions weakens the quality of the research.
Author`s response:
We too find it unfortunate that two interventions demonstrated a lack of feasibility. We do believe that, for the sake of transparency, we should report and share research findings even if they are not optimal.
- The paper could focus on stigma and early detection and treatment of psychological distress which is very relevant and could be explored with more detailed analysis.
Author`s response:

Reviewer 3 Report
Thanks for the opportunity to review this paper. Overall I found it to be of some interest, though the results are mainly negative, e.g. no participant was interested in undertaking directly administered antiretroviral therapy, and it was also difficult to recruit participants for the group medical appointment arm.
With regards to your conclusions, I do wonder if directly administered therapy may have been more acceptable if this was carried out in the person's home, thus obviating the need for the person to attend the clinic several times per week?
In addition, the group medical appointment seemed to have some benefits, though only for a small number of people. Do you think this could still be offered, knowing that most will not accept, though a small number may in fact derive benefit from it?
Author Response
Concerning: Revision manuscript: "Feasibility of four interventions to improve treatment adherence in migrants living with HIV in the Netherlands"
Dear Ms. Tomovic,
We would like to thank the reviewers for their time and useful contribution that has helped to improve the manuscript. In addition, we are pleased to read that reviewer 2 mentions that the subject of our manuscript and its study population are considered as highly relevant for care providers managing people living with HIV.
Hereby, we would like to submit our revised manuscript for publication as an original article in Diagnostics after carefully evaluating the comments of reviewers 1, 2, 3, and revising our manuscript accordingly. We will respond point-by-point to the reviewers` comments.
Replies to the comments of reviewer 3.
Author`s response:
As the participants of this intervention did appreciate the intervention, we believe that it has potential to benefit people living with HIV. However, we also conclude that the feasibility of this intervention is limited at this time as fear of HIV-related stigma resulting from being recognized by other GMA participants, and fear of third party disclosure appear to be key barriers to participation in GMAs for MLWH. These factors are even more present in MLWH due to specific migration related issues. HIV-related stigma should first be addressed before GMAs can be used to improve treatment adherence among MLWH. This was also described in the ‘Discussio; (page 11-12, lines 307-316).
Additionally, for an intervention to be feasible, the benefits need to be such that the investment outweighs the costs in terms of time, human resources and finances.
Assesment of English language and style
Author 6 (S.E. Stutterheim), is a native English speaker and has reviewed the manuscript.
Sincerely,
Prof.dr. A. Verbon

Round 2
Reviewer 2 Report
Thanks for acknowledging the limitations of the study and proposing corrective measures in your revised manuscript. Your study findings will contribute to the knowledge of conducting health interventions among migrant populations specifically against highly stigmatizing diseases like HIV.
Author Response
Could you explain in more detail the reasons why the ethics committee excludes the need for approval of the study? Lines 89-90 "The Medical Ethics Committee of the Erasmus University Medical Center and Maasstad Hospital have granted this study exemption from ethical approval (reference number: MEC-2012-399)".
Reply to Academic Editor
In the Netherlands research is divided in need for ethical approval for studies which invade physical integrity (such as blood draws or other fluid/organ sampling) just for study sake and for research involving new therapies. This kind of research has to be judged by local and national ethical committees and has to follow the rules set by WMO (wet medisch onderzoek =law medical research) legislegtion. Research not invading physical integrity or evaluating new therapies is deemed niet-WMO (= not requiring to fulfill legislation which is needed for WMO research). Our study was reviewed by the medical ethical committees of both hospitals and judged to be niet-WMO. So our study has been judged by the ethical committees, the required written informed consent of the patients and patient information have been approved by the ethical committees.
We used the phrase exemption for this meaning, the research was niet-WMO and was exempted from WMO legislation but stil had to fulfill the ethical requirements for human research. We have used this phrasing for other manuscripts in other medical journals.
However, if another phrasing is needed, this would be: "The Medical Ethics Committee of the Erasmus University Medical Center and Maasstad Hospital haveevaluated and approved the study (reference number: MEC-2012-399)".